# Harmonic Prior Self-conditioned Flow Matching for Multi-Ligand Docking and Binding Site Design

## Abstract

A significant amount of protein function requires binding small molecules, including enzymatic catalysis. As such, designing binding pockets for small molecules has several impactful applications ranging from drug synthesis to energy storage. Towards this goal, we first develop HARMONICFLOW, an improved generative process over 3D protein-ligand binding structures based on our self-conditioned flow matching objective. FLOWSITE extends this flow model to jointly generate a protein pocket's discrete residue types and the molecule's binding 3D structure. We show that HARMONICFLOW improves upon the state-of-the-art generative processes for docking in simplicity, generality, and performance. Enabled by this structure model, FLOWSITE designs binding sites substantially better than baseline approaches and provides the first general solution for binding site design.

## 1 Introduction

Designing proteins that can bind small molecules has many applications, ranging from drug synthesis to energy storage or gene editing. Indeed, a key part of any protein's function derives from its ability to bind and interact with other molecular species. For example, we may design proteins that act as antidotes that sequester toxins or design enzymes that enable chemical reactions through catalysis, which plays a major role in most biological processes. We develop FLOWSITE to address this design challenge by building on recent advances in deep learning (DL) based protein design [Dauparas et al., 2022] and protein-molecule docking [Corso et al., 2023].

Specifically, we aim to design protein pockets to bind a certain small molecule (called ligand). We assume that we are given a protein pocket via the 3D backbone atom locations of its residues as well as the 2D chemical graph of the ligand. We do not assume any knowledge of the 3D structure or the binding pose of the ligand. Based on this information, our goal is to predict the amino acid identities for the given backbone locations (see Figure 1). We also consider the more challenging task of designing

Figure 1: **Binding site design.** Given the backbone (green) and multi-ligand without structure, FLOWSITE generates residue types and structure (orange) to bind the multi-ligand and its jointly generated structure (blue). The majority of the pocket is omitted for visibility.

pockets that simultaneously bind multiple molecules and ions (which we call multi-ligand). Such multi-ligand binding proteins are important, for example, in enzyme design, where the ligands correspond to reactants.

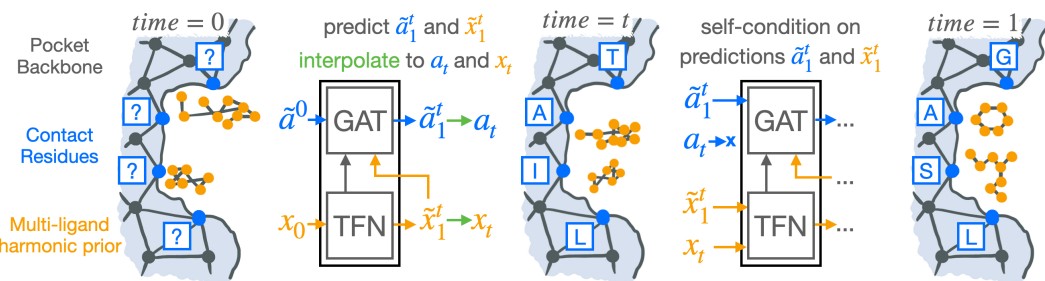

Figure 2: **Overview of FlowSite.** The generative process starts from a protein pocket's backbone atoms, initial residue types $\tilde{a}^0$, and initial ligand positions $x_0$. Our joint discrete-continuous self-conditioned flow updates them to $a_t$, $x_t$ by following its vector field defined by the model outputs $\tilde{a}_1^t$, $\tilde{x}_1^t$. This integration is repeated until reaching $time = 1$ with the produced sample $a_1$, $x_1$.

This task has not been addressed by deep learning yet. While deep learning has been successful in designing proteins that can bind to other proteins [Watson et al., 2023], designing (multi-)ligand binders is different and arguably harder in various aspects. For example, no evolutionary information is directly available, unlike when modeling interactions between amino acids only. The existing approaches, such as designing 6 drug binding proteins Polizzi & DeGrado [2020] or a single enzyme Yeh et al. [2023], build on expert knowledge and require manual steps. Therefore, we develop FLOWSITE as a more general and automated approach and the first deep learning solution for designing pockets that bind small molecules.

FLOWSITE is a flow-based generative model over discrete (residue identities) and continuous (ligand pose) variables. Our flow matching training criterion guides the model to learn a self-conditioned flow that jointly generates the contact residues and the (multi-)ligand 3D binding pose structures. To achieve this, we first develop HARMONICFLOW as a suitable generative process for 3D poses of (multi-)ligands. FLOWSITE extends this process to residue types. Starting from initial residue types and ligand atom locations sampled from a harmonic prior FLOWSITE updates them by iteratively following the learned vector field, as illustrated in Figure 2.

The HARMONICFLOW component of FLOWSITE performs the task known as docking, i.e., it realizes the 3D binding pose of the multi-ligand. As a method, it is remarkably simple in comparison to existing generative processes for docking, including the state-of-the-art diffusion process of DIFF-DOCK [Corso et al., 2023] that operates on ligand's center of mass, orientation, and torsion angles. HARMONICFLOW simply updates the cartesian coordinates of the atoms, yet manages to produce chemically plausible molecular structures without restricting ligand flexibility to torsions. Moreover, HARMONICFLOW outperforms DIFFDOCK's diffusion in multiple new pocket-level docking tasks on PDBBind. For instance, HARMONICFLOW achieves 24.4% of its predictions to be within root-mean-square-distance (RMSD) below 2Å as opposed to 16.3% for DIFFDOCK's diffusion.

Having established HARMONICFLOW as an improved generative process over ligand positions, we extend it to include discrete residue types to obtain FLOWSITE. We also adopt an additional "fake-ligand" data augmentation step where side chains are treated as ligands in order to realize additional training cases. Altogether, FLOWSITE is able to recover 47.0% of binding site amino acids compared to 39.4% of a baseline approach. This nearly closes the gap to an oracle method (51.4% recovery) with access to the ground truth 3D structure/pose of the ligand. Next to technical innovations such as self-conditioned flow matching or equivariant refinement TFN layers, our main contributions are:

1. The first application and investigation of flow matching for real-world biomolecular structure generation tasks and comparisons with diffusion model approaches.

2. FLOWSITE as the first deep learning solution to design binding sites for small molecules and a novel elegant framework to jointly generate discrete and continuous data.

3. HARMONICFLOW which improves upon the state-of-the-art generative process for generating 3D ligand binding structures in performance, simplicity, and applicability/generality.

## 2 Related Work

**Deep learning for Docking.** Designing binding sites with high affinity for a ligand requires reasoning about the binding free energy, which is deeply interlinked with modeling ligand binding 3D

structures. This task of molecular docking has recently been tackled with deep-learning approaches [Stärk et al., 2022; Lu et al., 2022; Zhang et al., 2023] including generative models [Corso et al., 2023; Qiao et al., 2023]. These generative methods are based on diffusion models, building on DIFF-DOCK [Corso et al., 2023], which combines diffusion processes over the ligand's torsion angles and position with respect to the protein. For the task of multi-ligand docking, no deep learning solutions exist yet, and we provide the first with HARMONICFLOW. We refer to Appendix D for additional important related work on this and the following topics.

**Protein Design.** A significant technical challenge for protein design is jointly modeling structure and sequence. Inverse folding approaches [Dauparas et al., 2022; Gao et al., 2023a; Yi et al., 2023; Hsu et al., 2022; Gao et al., 2023b] attempt this by designing new sequences given a protein structure. This is akin to our task where the protein pocket's backbone structure is given, and we aim to design its residue types to bind a (multi-)ligand. However, the only inverse folding method that models small molecules is Carbonara [Krapp et al., 2023], which is restricted to the 31 most common ligands of PDB and requires their 3D structure and position relative to the protein to be known. For general binding site design, this would not be the case, and predicting them with traditional docking methods would not be possible since they require the pocket side chain's 3D structure.

**Flow Matching.** This recent generative modeling paradigm [Lipman et al., 2022; Albergo & Vanden-Eijnden, 2022; Albergo et al., 2023] generalizes diffusion models [Ho et al., 2020; Song et al., 2021] in a simpler framework. Flow matching admits more design flexibility and multiple works [Tong et al., 2023b; Pooladian et al., 2023] showed how it enables learning flows between arbitrary start and end distributions in a simulation-free manner. It is easily extended to data on manifolds [Chen & Lipman, 2023] and comes with straighter paths that enable faster integration.

We provide the first applications of flow matching to real-world biomolecular tasks (multi-ligand docking and binding site design). While Klein et al. [2023] explored flow matching for 3D point clouds, their application was limited to overfitting on the Boltzmann distribution of a single molecule. We explain flow matching in Section 3.1.

# 3 Method

Our goal is to design binding pockets for a ligand where we assume the inputs to be the ligand's 2D chemical graph and the backbone coordinates of the pocket's residues. In this section, we lay out how FLOWSITE achieves this by first explaining our HARMONICFLOW generative process for docking in 3.1 before covering how FLOWSITE extends it to include discrete residue types in 3.2 and concluding with our model architecture in 3.3.

**Overview and definitions.** As visualized in Figure 2, FLOWSITE is a flow-based generative model that jointly updates discrete residue types and continuous ligand positions. The inputs are a protein pocket's backbone atoms $y \in \mathbb{R}^{L \times 4 \times 3}$ for $L$ residues with 4 atoms each and the chemical graph of a (multi-)ligand. Based on the ligand connectivity, its initial coordinates $x \in \mathbb{R}^{n \times 3}$ are sampled from a harmonic prior, and we initialize residue types $a \in \{1, \ldots, 20\}^L$ with an initial token (we drop the chemical information of the ligands in our notation for brevity).

Given this at time $t = 0$, the flow model $v_\theta$ with learned parameters $\theta$ iteratively updates residue types and ligand coordinates by integrating the ODE it defines. These integration steps are repeated from time $t = 0$ to time $t = 1$ to obtain the final generated binding pocket designs.

## 3.1 HarmonicFlow Structure Generation

We first lay out HARMONICFLOW for pure structure generation without residue type estimation. Our notation drops $v_\theta$'s conditioning on the pocket $y$ and residue estimates $a$ in this subsection (see the Architecture Section 3.3 for how $y$ is included). Simply put, HARMONICFLOW is flow matching with a harmonic prior, self-conditioning, and $x_1$ prediction (our refinement TFN layers in Section 3.3 are also important for performance). In more detail:

**Conditional Flow Matching.** Given the data distribution $p_1$ of bound ligand structures and any easy-to-sample prior $p_0$ over $\mathbb{R}^{n \times 3}$, we wish to learn an ODE that pushes the prior forward to the data distribution when integrating it from time 0 to time 1. The ODE will be defined by a time-

dependent vector field $v_\theta(\cdot, \cdot) : \mathbb{R}^{n \times 3} \times [0, 1] \mapsto \mathbb{R}^{n \times 3}$. Starting with a sample $\boldsymbol{x}_0 \sim p_0(\boldsymbol{x}_0)$ and following/integrating $v$ through time will produce a sample from the data distribution $p_1$.

To see how to train $v_\theta$, let us first assume access to a time-dependent vector field $u_t(\cdot)$ that would lead to an ODE that pushes from the prior $p_0$ to the data $p_1$ (it is not straightforward how to construct this $u_t$). This gives rise to a probability path $p_t$ by integrating $u_t$ until time $t$. If we could sample $\boldsymbol{x} \sim p_t(\boldsymbol{x})$ we could train $v_\theta$ with the unconditional flow matching objective [Lipman et al., 2022]

$$\mathcal{L}_{FM} = \mathbb{E}_{t \sim \mathcal{U}[0,1], \boldsymbol{x} \sim p_t(\boldsymbol{x})} \|v_\theta(\boldsymbol{x}, t) - u(\boldsymbol{x}, t)\|^2. \tag{1}$$

Among others, Tong et al. [2023b] show that to construct such a $u_t$ (that transports from prior $p_0$ to $p_1$), we can use samples from the data $\boldsymbol{x}_1 \sim p_1(\boldsymbol{x}_1)$ and prior $\boldsymbol{x}_0 \sim p_0(\boldsymbol{x}_0)$ and define $u_t$ via

$$u_t(\boldsymbol{x}) = \mathbb{E}_{\boldsymbol{x}_1 \sim p_1(\boldsymbol{x}_1), \boldsymbol{x}_0 \sim p_0(\boldsymbol{x}_0)} \frac{u_t(\boldsymbol{x}|\boldsymbol{x}_0, \boldsymbol{x}_1) p_t(\boldsymbol{x}|\boldsymbol{x}_0, \boldsymbol{x}_1)}{p_t(\boldsymbol{x})} \tag{2}$$

where we can choose easy-to-sample conditional flows $p_t(\cdot|\cdot, \cdot)$ that give rise to simple conditional vector fields $u_t(\cdot|\cdot, \cdot)$. We still cannot efficiently compute this $u_t(\boldsymbol{x})$ and use it in $\mathcal{L}_{FM}$ because we do not know $p_t(\boldsymbol{x})$, but there is no need to: it is equivalent to instead train with the following conditional flow matching loss since $\nabla_\theta \mathcal{L}_{FM} = \nabla_\theta \mathcal{L}_{CFM}$.

$$\mathcal{L}_{CFM} = \mathbb{E}_{t \sim \mathcal{U}[0,1], \boldsymbol{x}_1 \sim p_1(\boldsymbol{x}_1), \boldsymbol{x}_0 \sim p_0(\boldsymbol{x}_0), x \sim p_t(\boldsymbol{x}|\boldsymbol{x}_0, \boldsymbol{x}_1)} \|v_\theta(\boldsymbol{x}, t) - u_t(\boldsymbol{x}|\boldsymbol{x}_0, \boldsymbol{x}_1)\|^2. \tag{3}$$

Our simple choice of conditional probability path is $p_t(\boldsymbol{x}|\boldsymbol{x}_0, \boldsymbol{x}_1) = \mathcal{N}(\boldsymbol{x}|t\boldsymbol{x}_1 + (1-t)\boldsymbol{x}_0, \sigma^2)$, which gives rise to the conditional vector field $u_t(\boldsymbol{x}|\boldsymbol{x}_0, \boldsymbol{x}_1) = \boldsymbol{x}_1 - \boldsymbol{x}_0$. Notably, we find it helpful to parameterize $v_\theta$ to predict $\boldsymbol{x}_1$ instead of $(\boldsymbol{x}_1 - \boldsymbol{x}_0)$.

*Training* with the conditional flow matching loss then boils down to 1) Sample data $\boldsymbol{x}_1 \sim p_1(\boldsymbol{x}_1)$ and prior $\boldsymbol{x}_0 \sim p_0(\boldsymbol{x}_0)$. 2) Interpolate between between the points. 3) Add noise to the interpolation to obtain $x$. 4) Evaluate and minimize $\mathcal{L}_{CFM} = \|v_\theta(\boldsymbol{x}, t) - \boldsymbol{x}_1\|^2$ with it. *Inference* is just as straightforward. We sample from the prior $\boldsymbol{x}_0 \sim p_0(\boldsymbol{x}_0)$ and integrate from $t = 0$ to $t = 1$ with an arbitrary ODE solver. We use an Euler solver, i.e., we iteratively predict $\boldsymbol{x}_1$ as $\tilde{\boldsymbol{x}}_1 = v_\theta(\boldsymbol{x}_t, t)$, and then calculate the step size scaled velocity estimate from it and add it to the current point $\boldsymbol{x}_{t+\Delta t} = \boldsymbol{x}_t + \Delta t(\tilde{\boldsymbol{x}}_1 - \boldsymbol{x}_0)$. Training and inference algorithms are in Appendix A.4.

**Harmonic Prior.** Any prior can be used for $p_0$ in the flow matching framework. We choose a harmonic prior as in Eigen-Fold [Jing et al., 2023] that samples atoms to be close to each other if they are connected by a bond. We identify this as an especially valuable inductive bias when dealing with multiple molecules and ions since atoms of different molecules are already spatially separated at $t = 0$ as visualized in Figure 3.

This prior is constructed based on covalent bonds that define a graph with adjacency matrix $\boldsymbol{A}$ from which we can construct the graph Laplacian $\boldsymbol{L} = \boldsymbol{D} - \boldsymbol{A}$ where $\boldsymbol{D}$ is

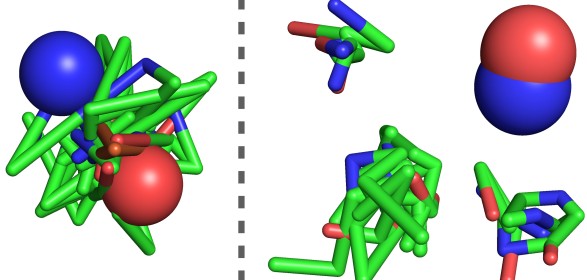

Figure 3: **Harmonic Prior.** Initial positions for the same single multi-ligand from an isotropic Gaussian (*left*) and from a harmonic prior (*right*). (Bound structure for this multi-ligand is in Figure 1).

the degree matrix. The harmonic prior is then $p_0(\boldsymbol{x}_0) \propto exp(-\frac{1}{2}\boldsymbol{x}_0^T \boldsymbol{L} \boldsymbol{x}_0)$ which can be sampled as a transformed gaussian.

**Structure Self-conditioning.** With this, we aim to bring AlphaFold2's [Jumper et al., 2021] successful recycling strategy to flow models for structure generation. Recycling enables training a deeper structure predictor without additional memory cost by performing multiple forward passes while only computing gradients for the last. For flow matching, we achieve the same by adapting the discrete diffusion model self-conditioning approach of Chen et al. [2023] as follows:

Instead of defining the vector field $v_\theta(\boldsymbol{x}_t, t)$ as a function of $\boldsymbol{x}_t$ and $t$ alone, we additionally condition it on the prediction $\tilde{\boldsymbol{x}}_1^t$ of the previous integration step and use $v_\theta(\boldsymbol{x}_t, \tilde{\boldsymbol{x}}_1^t, t)$. At the beginning

of *inference* the self-conditioning input is a sample from the harmonic prior $\tilde{\boldsymbol{x}}_1^0 \sim p_0(\tilde{\boldsymbol{x}}_1^0)$. In all following steps, it is the flow model's output (its prediction of $\boldsymbol{x}_1$) of the previous step $\tilde{\boldsymbol{x}}_1^t = v_\theta(\boldsymbol{x}_{t-\Delta t}, \tilde{\boldsymbol{x}}_1^{t-\Delta t}, t - \Delta t)$. To *train* this, in a random 50% of the training steps, the self-conditioning input is a sample from the prior $\tilde{\boldsymbol{x}}_1^0$. In the other 50%, we first generate a self-conditioning input $\tilde{\boldsymbol{x}}_1^{t+\Delta t} = v_\theta(\boldsymbol{x}_t, \tilde{\boldsymbol{x}}_1^0, t)$, detach it from the gradient computation graph, and then use $v_\theta(\boldsymbol{x}_t, \tilde{\boldsymbol{x}}_1^{t+\Delta t}, t)$ for the loss computation. Algorithms 3 and 4 show these training and inference procedures.

## 3.2 FlowSite Binding Site Design

In the FLOWSITE binding site design framework, HARMONICFLOW $\tilde{\boldsymbol{x}}_1^{t+\Delta t} = v_\theta(\boldsymbol{x}_t, \tilde{\boldsymbol{x}}_1^t, t)$ is augmented with an additional self-conditioned flow over the residue types to obtain $(\tilde{\boldsymbol{x}}_1^{t+\Delta t}, \tilde{\boldsymbol{a}}_1^{t+\Delta t}) = v_\theta(\boldsymbol{x}_t, \tilde{\boldsymbol{x}}_1^t, \boldsymbol{a}_t, \tilde{\boldsymbol{a}}_1^t, t)$. The flow no longer produces $\tilde{\boldsymbol{x}}_1^{t+\Delta t}$ as an estimate of $\boldsymbol{x}_1$ and then interpolates to $\boldsymbol{x}_{t+\Delta t}$ but instead produces $(\tilde{\boldsymbol{x}}_1^{t+\Delta t}, \tilde{\boldsymbol{a}}_1^{t+\Delta t})$ from which we obtain the interpolation $(\boldsymbol{x}_{t+\Delta t}, \boldsymbol{a}_{t+\Delta t})$ and use it for the next integration step (see Figure 4). The start $\boldsymbol{a}_0, \tilde{\boldsymbol{a}}_1^0$ are initialized as a mask token while the structures $\boldsymbol{x}_0, \tilde{\boldsymbol{x}}_1^0$ are drawn from a harmonic prior.

This joint discrete-continuous data flow is trained with the same self-conditioning strategy as in structure self-conditioning, but with the additional discrete self-conditioning input $\tilde{\boldsymbol{a}}_1^1$ that is either a model output or a mask token. To the training loss we add the cross-entropy $\mathcal{L}_{type}$ between $\boldsymbol{a}$ and $\tilde{\boldsymbol{a}}_1^t$. In practice, we find that the $\boldsymbol{a}_1$ prediction $\tilde{\boldsymbol{a}}_1^t$ already carries most information that is useful for predicting $a_1$ and we omit the interpolation $\boldsymbol{a}_t$ as model input to obtain the simpler $(\tilde{\boldsymbol{x}}_1^{t+\Delta t}, \tilde{\boldsymbol{a}}_1^{t+\Delta t}) = v_\theta(\boldsymbol{x}_t, \tilde{\boldsymbol{x}}_1^t, \tilde{\boldsymbol{a}}_1^t, t)$. This formulation admits a direct interpretation as recycling Jumper et al. [2021] and a clean joint discrete-continuous process without defining a discrete data interpolation.

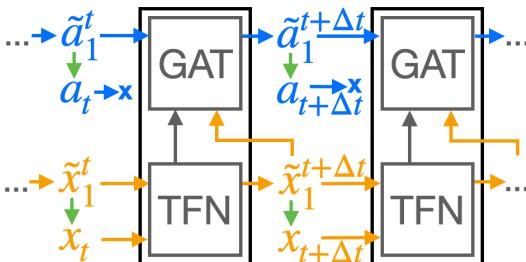

Figure 4: **FlowSite self-conditioned updates.** Residue type predictions $\tilde{\boldsymbol{a}}_1^t$ from invariant GAT layers and position predictions $\tilde{\boldsymbol{x}}_1^t$ from equivariant TFN layers are used as self-conditioning inputs and to interpolate to the updates $\boldsymbol{a}_t, \boldsymbol{x}_t$.

**Fake Ligand Data Augmentation.** This strategy is based on the evidence of Polizzi & DeGrado [2020] that a protein's sidechain-sidechain interactions are similar to sidechain-ligand interactions for tight binding. In our optional data augmentation, we train with 20% of the samples having a *"fake ligand"*. Given a protein, we construct a fake ligand as the atoms of a randomly selected residue that has at least 4 other residues within 4Å heavy atom distance. Additionally, we modify the protein by removing the residue that was chosen as the fake ligand and the residues that are within 7 positions next to that residue in the protein chain, as visualized in Figure 5.

## 3.3 Architecture

Here, we provide an overview of the FLOWSITE architecture (visualized in Appendix Figure 6) that outputs ligand positions $\tilde{\boldsymbol{x}}_1$ and uses them for a residue type prediction $\tilde{\boldsymbol{a}}_1$. The structure prediction is produced by a stack of our SE(3)-equivariant refinement TFN layers that are crucial for the performance of HARMONICFLOW's structure generation. This is followed by invariant layers to predict the invariant residue types. The precise architecture definition is in Appendix A.6 and an architecture visualization in Figure 6.

**Radius Graph Representation.** We represent the (multi-)ligand and the protein as graphs where nodes are connected based on their distances. Each protein residue and each ligand atom is a node. These are connected by protein-to-protein edges, ligand-to-ligand edges, and edges between ligand and protein. While only a single node is assigned to each residue, they contain information about all backbone atom positions (N, Ca, C, O).

**Equivariant refinement Tensor Field Network (TFN) layers.** Based on TFN [Geiger et al., 2020], these layers capture the important inductive bias of SE(3)-equivariance (translating and rotating the input will equally translate and rotate the output). They are a remarkably simple yet effective tweak from previous message passing TFNs [Jing et al., 2022; Corso et al., 2023], where we instead update and refine ligand coordinates with each layer akin to EGNNs [Hoogeboom et al., 2022].

The $k$-th refinement TFN layer takes as input the protein positions $\boldsymbol{y}$, current ligand positions $\boldsymbol{x}_t$, and features $\boldsymbol{h}^{k-1}$ (with $\boldsymbol{h}^0$ being zeros for the ligand and vectors between N, Ca, C, O for the protein). We construct equivariant messages for each edge via a tensor-product of neighboring nodes' invariant and equivariant features. The messages include the structure self-conditioning information by using the interatomic distances of the self-conditioning input $\boldsymbol{x}_1^t$ to parameterize the tensor products. We sum the messages to obtain new node features $\boldsymbol{h}^{k+1}$ and use them as input to an O(3) equivariant linear layer to predict intermediate refined ligand coordinates $\hat{\boldsymbol{x}}_1^k$. Before passing $\hat{\boldsymbol{x}}_1^k$ to the next refinement TFN layer, we detach them from the gradient computation graph for the non-differentiable radius graph building of the next layer.

After a stack of $K$ TFN refinement layers, the positions $\hat{\boldsymbol{x}}_1^K$ are used as final prediction $\tilde{\boldsymbol{x}}_1^{t+\Delta t}$. While $\tilde{\boldsymbol{x}}_1^{t+\Delta t}$ is supervised with the conditional flow matching loss $\mathcal{L}_{CFM} = \|\tilde{\boldsymbol{x}}_1^{t+\Delta t} - \boldsymbol{x}_1\|^2$ the intermediate positions $\hat{\boldsymbol{x}}_1^k$ contribute to an additional refinement loss $\mathcal{L}_{refine} = \sum_{k=1}^{K-1} \|\hat{\boldsymbol{x}}_1^k - \boldsymbol{x}_1\|^2$.

**Invariant Network.** The inputs to this part of FLOWSITE are the TFN's ligand structure prediction $\tilde{\boldsymbol{x}}_1$, the protein structure $\boldsymbol{y}$, the invariant scalar features of the refinement TFN layers, and the self-conditioning input $\boldsymbol{a}_1^t$. From the protein structure, we construct on PiFold's [Gao et al., 2023a] distance-based invariant edge features and node features that encode the geometry of the backbone. For the edges between protein and ligand, we construct features that encode the distances from a ligand atom to all 4 backbone atoms of a connected residue.

These are processed by a stack of graph attention layers that update ligand and protein node features as well as edge features for each type of edge (ligand-to-ligand, protein-to-protein, and between the molecules). For each edge, the convolutional layers first predict attention weights from the edge features and the features of the nodes they connect. We then update a node's features by summing messages from each incoming edge weighted by the attention weights. Then, we update an edge's features based on its nodes' new features. A precise definition is in Appendix A.6. From the residue features after a stack of these convolutions, we predict new residue types $\boldsymbol{a}_{t+\Delta t}$ together with side chain torsion angles $\alpha$. We use those in an auxiliary loss $\mathcal{L}_{torsion}$ defined as in AlphaFold2's Appendix 1.9.1 [Jumper et al., 2021]. Thus, the complete loss for FLOWSITE is a weighted sum of $\mathcal{L}_{CFM}, \mathcal{L}_{refine}, \mathcal{L}_{type}$, and $\mathcal{L}_{torsion}$, while HARMONICFLOW only uses $\mathcal{L}_{CFM}$ and $\mathcal{L}_{refine}$.

# 4  Experiments

We evaluate FLOWSITE with the PDBBind and Binding MOAD datasets detailed in Appendix F. Every reported number is averaged over 10 generated samples for each ligand. Precise experimental details are in Appendix E and code to reproduce each experiment is at https://anonymous. 4open.science/r/wolf.

## 4.1  Question: HarmonicFlow Structure Generation Capability

Here, we consider the HARMONICFLOW component of FLOWSITE and investigate its binding structure generation capability. This is to find out whether HARMONICFLOW is fit for binder design where good structure generation is necessary for taking the 3D structure of the bound ligand into account. Additionally, we aim to determine how HARMONICFLOW compares with state-of-the-art structure generation and if its use for docking should be further explored.

**Task Setup.** This subsection only uses the HARMONICFLOW component of FLOWSITE - the architecture only contains refinement TFN layers, and there is no sequence prediction. The inputs are the (multi-)ligand's chemical graph and the protein pocket's backbone atoms and residue types (see Appendix Table 6 for experiments without residue type inputs). From this, the binding structure of the (multi-)ligand has to be inferred. There is also no *fake ligand data augmentation*. And we perform docking to *Distance-Pockets* and *Radius-Pockets* as described in Appendix A.1 and we provide preliminary blind docking results in Appendix C

**Baseline.** We compare with the state-of-the-art diffusion process of DIFFDOCK [Corso et al., 2023]. *Note that this is not the full DIFFDOCK docking pipeline:* Both HARMONICFLOW and DIFFDOCK's diffusion can generate multiple samples and, for the task of docking, a further discriminator (called confidence model in DIFFDOCK) could be used to select the most likely poses. We only compare the

Table 1: **HARMONICFLOW vs. DIFFDOCK DIFFUSION.** Comparison on PDBBind splits for docking into *Distance-Pockets* (residues close to ligand) and *Radius-Pockets* (residues within a radius of the pocket center). The columns "%<2" show the fraction of predictions with an RMSD to the ground truth that is less than 2Å (higher is better). "Med." is the median RMSD (lower is better).

| | Sequence Similarity Split | | | | Time Split | | | |
| | Distance-Pocket | | Radius-Pocket | | Distance-Pocket | | Radius-Pocket | |
| Method | %<2 | Med. | %<2 | Med. | %<2 | Med. | %<2 | Med. |
|---|---|---|---|---|---|---|---|---|
| DIFFDOCK DIFFUSION | 28.4 | 3.1 | 16.3 | 3.9 | 26.6 | 3.2 | 15.5 | 4.1 |
| HARMONICFLOW | 31.8 | 3.0 | 24.4 | 3.2 | 45.9 | 2.3 | 37.8 | 2.7 |

3D structure generative models and neither use language model residue embeddings. See Appendix E for details on retraining DIFFDOCK.

**PDBBind docking results.** In Table 1, we find that our flow matching based HARMONICFLOW outperforms DIFFDOCK's diffusion in producing ligand structures close to the ground truth for both splits of PDBBind. This shows that DIFFDOCK's restriction of the generative process to the lower dimensional manifold of rotations, torsions, and translations is not necessary. Flow matching's straighter paths, along with our well-chosen prior and self-conditioning, can achieve better performance (we investigate flow matching further in Section 4.3). Furthermore, the sampled conformations in Figure 7 and videos of the generation process show that HARMONICFLOW produces chemically plausible structures and well captures the physical constraints of interatomic interactions.

**Binding MOAD multi-ligand docking results.** For binding site design, it is often necessary to model multiple ligands and ions (e.g., reactants for an enzyme). We test this with Binding MOAD, which contains such multi-ligands. Since no deep learning solutions for multi-ligands exist yet and traditional docking methods would require side-chain atom locations, we compare with EIGENFOLD's [Jing et al., 2023] Diffusion

Table 2: **Multi-Ligand Docking.** Structure generation performance on Binding MOAD's *multi-ligands*. "%<2" means the fraction of predictions with an RMSD to the ground truth less than 2Å (higher better). "Med." is the median RMSD (lower better).

| Method | %<2 | %<5 | Med. |
|---|---|---|---|
| EIGENFOLD DIFFUSION | 39.7 | 73.5 | 2.4 |
| HARMONICFLOW | 44.4 | 75.0 | 2.2 |

and provide qualitative evaluation in Appendix Figure 7. For EIGENFOLD DIFFUSION, we use the same model as HARMONICFLOW, including its improved coordinate update layers and predict $x_0$ (in what corresponds to $x_0$ prediction in diffusion models), which we found to work better. Table 2 shows HARMONICFLOW as viable for docking multi-ligands - thus, the first ML method for this task with important applications besides binding site design.

## 4.2 Question 2: Binding Site Recovery

**Setup.** The input to FLOWSITE is the binding pocket/site specified by its backbone and the chemical identity of the ligand (without its 3D structure). We use two metrics, sequence recovery (percentage of correctly predicted residues) and our new residue similarity aware *BLOSUM score* defined in Appendix A.2.

**Baselines.** PIFOLD *(no ligand)* is the architecture of Gao et al. [2023a] and does not use any ligand information. In PIFOLD *(2D ligand)*, we first process the ligand with PNA [Corso et al., 2020] message passing and pass its features as additional input to the PIFOLD architecture. Lastly, GROUND TRUTH POS and RANDOM LIGAND POS use the architecture of FLOWSITE without the ligand structure prediction layers. Instead, the ligand positions are either the ground truth bound structure or sampled from a standard Normal at the pocket's alpha carbon center of mass. The oracle GROUND TRUTH POS method also uses fake ligand data augmentation.

**Pocket Recovery Results.** Table 3 shows that FLOWSITE consistently is able to recover the original pocket better than simpler treatments of the (multi-)ligand, closing the gap to the oracle method that has access to the ground truth ligand structure. The joint structure generation helps in determining the original residue types (keeping in mind that these are not necessarily the only or best). RANDOM

Table 3: **Binding Site Recovery.** Comparison on PDBBind and Binding MOAD sequence similarity splits for recovering residues of binding sites. *Recovery* is the percentage of correctly predicted residues, and *BLOSUM score* takes residue similarity into account. 2D ligand refers to a simple GNN encoding of the ligand's chemical graph as additional input. The GROUND TRUTH POS row has access to the, in practice, unknown ground truth 3D crystal structure of the ligand and protein.

| Method | Binding MOAD | | PDBBind | |
|---|---|---|---|---|
| | *BLOSUM score* | *Recovery* | *BLOSUM score* | *Recovery* |
| PIFOLD (no ligand) | 35.2 | 39.4 | 40.7 | 43.5 |
| PIFOLD (2D ligand) | 35.7 | 40.4 | 42.2 | 44.5 |
| RANDOM LIGAND POS | 38.2 | 41.8 | 41.5 | 43.7 |
| FLOWSITE | 44.3 | 47.0 | 47.1 | 48.5 |
| GROUND TRUTH POS | 48.4 | 51.4 | 51.3 | 51.2 |

LIGAND POS further confirms that inferring approximate ligand coordinates, like HARMONICFLOW in FLOWSITE, is crucial for recovering the binding pocket.

### 4.3 Question 3: Ablations and Flow-Matching Investigation

**Investigations.** EIGENFOLD DIFFUSION, as described in 4.1 is an adaption of Jing et al. [2022]'s diffusion process. This essentially replaces the flow matching based generative process of HARMONICFLOW with a diffusion process. In VELOCITY PREDICTION, the TFN model predicts $(x_1 - x_0)$ instead of $x_1$ meaning that $\mathcal{L}_{CFM} = \|v_\theta - (x_1 - x_0)\|^2$. In STANDARD TFN LAYERS, our refinement TFN layers are replaced, meaning that there are no intermediate position updates - only the last layer produces an update. NO SELF-CONDITIONING does not use our structure

Table 4: **Flow matching investigation.** Variations of flow matching, diffusion, and architecture choices compared with our HARMONICFLOW on a PDBBind sequence similarity split with *Radius-Pockets*.

| Method | %<2 | %<5 | Med. |
|---|---|---|---|
| EIGENFOLD DIFFUSION | 21.0 | 65.2 | 3.8 |
| VELOCITY PREDICTION | 16.4 | 64.6 | 3.7 |
| STANDARD TFN LAYERS | 16.6 | 71.9 | 3.4 |
| NO SELF-CONDITIONING | 20.7 | 69.3 | 3.4 |
| HARMONICFLOW $\sigma = 0$ | 25.4 | 69.9 | 3.2 |
| HARMONICFLOW $\sigma = 0.5$ | 24.4 | 69.8 | 3.2 |

self-conditioning. SIGMA=0 uses $\sigma = 0$ for the conditional flow, corresponding to a deterministic interpolant for training.

**Results.** Table 4 shows the importance of our self-conditioned flow matching objective, which enables refinement of the binding structure prediction $\tilde{x}_1^t$ next to updates of $x_t$ at little additional training time - a 12.8% increase in this experiment. Furthermore, the refinement TFN layers improve structure prediction substantially. Lastly, parameterizing the vector field to predict $x_1$ instead of $(x_1 - x_0)$ appears more suitable for flow matching applications in molecular structure generation.

## 5 Conclusion

We proposed the HARMONICFLOW generative process for binding structure generation and FLOWSITE for binding site design. Our HARMONICFLOW improves upon the state-of-the-art generative process for docking in simplicity, applicability, and performance in various docking settings. We investigated how flow matching contributes to this, together with our technical innovations such as self-conditioned flow matching, harmonic prior ligands, or equivariant refinement TFN layers.

With FLOWSITE, we leverage our superior binding structure generative process and extend it to discrete residue types, resulting in a joint discrete-continuous flow model for designing ligand binding pockets. This is an important task for which FLOWSITE is the first general solution. FLOWSITE is a step toward binding site design, but recovery results cannot replace biological validation - this is future work we pursue. Additionally, we will address enzyme design by incorporating more prerequisites for catalytic activity besides binding the reactants.

## Author Contributions

Anonymized

## Acknowledgments

Anonymized

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

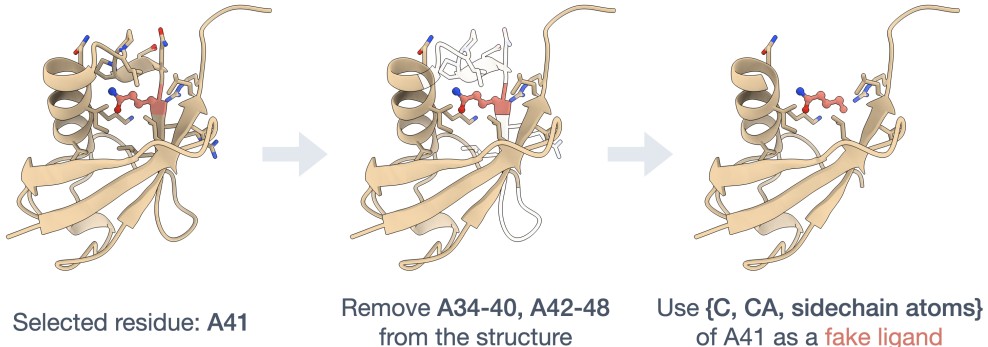



Selected residue: **A41**     Remove **A34-40, A42-48**     Use **{C, CA, sidechain atoms}**
from the structure     of A41 as a fake ligand



Figure 5: **Visualization of Fake Ligand creation.** Depicted is a fake ligand created for the Ubiquitin protein. Out of all residues that have at least 4 contacts with other residues (apart from those that are within 7 locations in the chain) a residue is randomly selected as the fake ligand. Then we remove the residue itself from the protein and all residues that are within 7 locations in the chain.

## A Method Details and Explanations

### A.1 Pocket Definitions

We test docking on the pocket level since that is the structure modeling capability required for the binding site design task (in Appendix C, we show preliminary results for docking to the whole protein). We define the binding pocket in two ways. The *Distance-Pocket* definition follows prior work [Méndez-Lucio et al., 2021a; Zhou et al., 2023] and includes any residue that has a heavy atom within 12 Å of any ground truth ligand heavy atom. This type of pocket might allow the models to reason where the ground truth ligand was based on which residues are included. Therefore, we additionally use *Radius-Pockets*: first, we select residues within 5 Å heavy atom distance of any ligand atom. The center of mass of these residues' alpha carbons is the pocket center. The final pocket includes all residues with an atom in a radius around the pocket center. This radius is the distance between the pocket center and the farthest ligand heavy atom plus 10Å.

### A.2 BLOSUM Score

Next to sequence recovery, we also evaluate with our *BLOSUM Score* in an attempt to penalize amino acid predictions less if the predicted residue type is similar yet different from the original residue. With $\boldsymbol{A} \in \mathbb{R}^{20 \times 20}$ being the BLOSUM62 matrix, $\boldsymbol{X} \in \mathbb{R}^{n \times 20}$ the one hot encoded ground truth residues types and $\hat{\boldsymbol{X}} \in \mathbb{R}^{n \times 20}$ the predicted residues types the *BLOSUM Score* is:

$$Score(\boldsymbol{X}, \hat{\boldsymbol{X}}) = \frac{\mathbf{1}^T diag(\boldsymbol{X} \boldsymbol{A} \hat{\boldsymbol{X}}^T)}{\mathbf{1}^T diag(\boldsymbol{X} \boldsymbol{A} \boldsymbol{X}^T)} \tag{4}$$

### A.3 Fake Ligand Data Augmentation Visualization

In Figure 5, we visualize the construction of our fake ligands as described in Section 3.2. When constructing the fake ligand from a residue, we drop the backbone oxygen and nitrogen of the amino acid and keep the carbon, alpha carbon, and the side chain as the ligand's atoms.

### A.4 Flow Matching Training and Inference

In Section 3.1, we lay out the conditional flow matching objective as introduced by Lipman et al. [2022] and extended to arbitrary start and end distributions by multiple works concurrently [Albergo & Vanden-Eijnden, 2022; Albergo et al., 2023; Pooladian et al., 2023; Tong et al., 2023b]. We presented conditional flow matching in this more general scenario where the prior $p_0$ and the data $p_1$ can be arbitrary distributions, as long as we can sample from the prior.

**Algorithm 1:** Conditional Flow Matching training with $x_1$ prediction and simple constant width gaussian conditional path.

**Input:** Training data distribution $p_1$, prior $p_0$, $\sigma$, and initialized vector field $v_\theta$
**while** *Training* **do**
    $x_0 \sim p_0(x_0); x_1 \sim p_1(x_1); t \sim \mathcal{U}(0,1);$
    $\mu_t \leftarrow tx_1 + (1-t)x_0;$
    $x \sim \mathcal{N}(\mu_t, \sigma^2 I);$
    $\mathcal{L}_{CFM} \leftarrow \|v_\theta(x,t) - x_1\|^2;$
    $\theta \leftarrow \text{Update}(\theta, \nabla_\theta \mathcal{L}_{CFM})$ ;
**return** $v_\theta$

---

**Algorithm 2:** Conditional Flow Matching inference with $x_1$ prediction and simple constant width gaussian conditional path.

**Input:** Prior $p_0$, number of integration steps T, and trained vector field $v_\theta$
$steps \leftarrow 1;$
$\Delta t \leftarrow 1/T;$
$t \leftarrow 0;$
$x_0 \sim p_0(x_0);$
$x_t \leftarrow x_0;$
**while** $steps \leq T - 1$ **do**
    $\tilde{x}_1 \leftarrow v_\theta(x_t, t)$ ;
    $x_t \leftarrow x_t + \Delta t(\tilde{x}_1 - x_0)$ ;
    $t \leftarrow t + \Delta t$ ;
**return** $x_t$

---

Many choices of conditional flows and conditional vector fields are possible. For different applications and scenarios, some choices perform better than others. We find it to already work well to use a very simple choice of conditional probability path $p_t(x|x_0, x_1) = \mathcal{N}(x|tx_1+(1-t)x_0, \sigma^2)$, which gives rise to the conditional vector field $u_t(x|x_0, x_1) = x_1 - x_0$. With this conditional flow and with parameterizing $v_\theta$ to predict $x_1$, the optimization and inference is remarkably straightforward as algorithms 1 and 2 show.

### A.5 Self-conditioned Flow Matching Training and Inference

In Section 3.1, we also explain the self-conditioning training and inference procedure. When additionally using self-conditioning, the training and inference algorithms are only slightly modified and still very simple as presented in algorithms 3 and 4.

### A.6 FLOWSITE Architecture

Here, we detail the FLOWSITE architecture as visualized in Figure 6 in more detail. The first half of the architecture is an equivariant Tensor Field Network [Thomas et al., 2018] while the second part is an invariant architecture with graph attention layers similar to the architecture of PIFOLD [Gao et al., 2023a] where edge features are also initialized and updated.

**Radius Graph.** The protein and (multi-)ligand are represented as graphs: each residue corresponds to a node, and each ligand atom is a node. Edges are drawn between residue nodes if they are within 50 Å, between ligand nodes if they are within 50 Å, and between the two molecules' nodes if they are within 30 Å. The locations of the residue nodes are given by their alpha carbons, while the atom locations provide the node positions for the ligand nodes.

**Node Features.** The ligand features as input to the TNF and to the invariant part of the architecture are atomic number; chirality; degree; formal charge; implicit valence; the number of connected hydrogens; hybridization type; whether or not it is in an aromatic ring; in how many rings it is; and finally, 6 features for whether or not it is in a ring of size 5 or 6.

**Algorithm 3:** Conditional Flow Matching training with $x_1$ prediction and simple constant width gaussian conditional path.

**Input:** Training data distribution $p_1$, prior $p_0$, $\sigma$, and initialized vector field $v_\theta$

**while** *Training* **do**

    $x_0 \sim p_0(x_0)$; $x_1 \sim p(x_1)$; $t \sim \mathcal{U}(0,1)$; $s \sim \mathcal{U}(0,1)$;

    $\mu_t \leftarrow tx_1 + (1-t)x_0$;

    $x \sim \mathcal{N}(\mu_t, \sigma^2 I)$;

    $\tilde{x}_1 \sim p_0(\tilde{x}_1)$;

    **if** $s > 0.5$ **then**

        $\tilde{x}_1 \leftarrow v_\theta(x, \tilde{x}_1 t)$);

    $\mathcal{L}_{CFM} \leftarrow \|v_\theta(x, \tilde{x}_1 t) - x_1\|^2$;

    $\theta \leftarrow \text{Update}(\theta, \nabla_\theta \mathcal{L}_{CFM})$ ;

**return** $v_\theta$

---

**Algorithm 4:** Conditional Flow Matching inference with $x_1$ prediction and simple constant width gaussian conditional path.

**Input:** Prior $p_0$, number of integration steps T, and trained vector field $v_\theta$

$steps \leftarrow 1$;

$\Delta t \leftarrow 1/T$;

$t \leftarrow 0$;

$\tilde{x}_1 \sim p(x_0)$;

$x_0 \sim p(x_0)$;

$x_t \leftarrow x_0$;

**while** $steps \leq T - 1$ **do**

    $\tilde{x}_1 \leftarrow v_\theta(x, \tilde{x}_1 t)$ ;

    $x_t \leftarrow x_t + \Delta t(\tilde{x}_1 - x_0)$ ;

    $t \leftarrow t + \Delta t$ ;

**return** $x_t$

---

The initial receptor features for the TFN are scalar feature encodings of the invariant residue types together with vector features, which are three vectors from the alpha carbon to N, C, and O.

For the invariant graph attention layer stack, the residue inputs are the invariant geometric encodings of PIFOLD [Gao et al., 2023a]. Additionally, they contain the residue type self-conditioning information via embeddings of the previously predicted features $\tilde{a}_1^t$ and the invariant scalar node features of the last refinement TFN layer.

Additionally, radial basis encodings of the sampling time $t$ of the conditional flow are added to all initial node features.

**Edge Features.** For the Tensor Field Network, the edge features are a radial basis embedding of the alpha carbon distances for the protein-to-protein edges, atom distances for the ligand-to-ligand edges, and alpha carbon to ligand atom distances for the edges between the protein and the ligand. Additionally, the ligand-to-ligand edges features obtain information of the structure self-conditioning by also adding the radial basis interatomic distance embeddings of the previously predicted ligand coordinates $\tilde{x}_1^t$ to them.

Meanwhile, for the invariant graph attention part of the architecture, the ligand-to-ligand edge features are only radial basis embeddings of the interatomic distances. The protein-to-protein edge features are given by radial basis encodings of all pairwise distances between the backbone atoms N, C, Ca, O, and an additional virtual atom (as introduced by PIFOLD) associated with each residue. The edges between the protein and ligand are featurized as the embeddings of the four possible distances between a single ligand atom and the four backbone atoms of a residue.

**Tensor Field Network.** The equivariant part of FLOWSITE uses our equivariant refinement TFN layers based on tensorfield networks [Thomas et al., 2018] and implemented using the `e3nn` library [Geiger et al., 2020]. These rely on tensor products between invariant and equivariant features. We

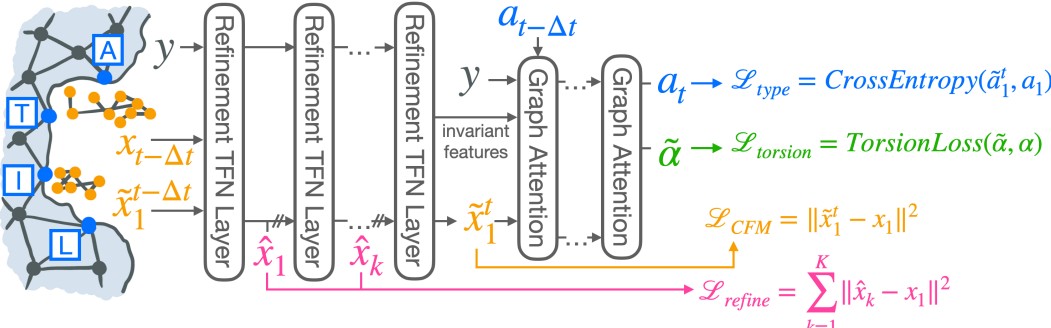

Figure 6: **FlowSite architecture.** The refinement TFN layers of HARMONICFLOW first update the ligand coordinates $\boldsymbol{x}_{t-\Delta t}$ multiple times to produce the structure prediction $\tilde{\boldsymbol{x}}_1^t$ from which $\tilde{\boldsymbol{x}}_1^t$ is computed. The TFN's invariant features and $\tilde{\boldsymbol{x}}_1$ are fed to invariant layers to produce side chain angles $\tilde{\alpha}$ as auxiliary training targets and the new residue estimate $\boldsymbol{a}_t$.

denote the tensor products as $\otimes_w$ where $w$ are the path weights. Further, we write the $i$-th node features after the $k$-th layer as $\boldsymbol{h}_i^k$ for the equivariant Tensorfield network layers. $\boldsymbol{h}_i^0$ is initialized as described above in the Node Features paragraph. Lastly, $\mathcal{N}_i$ denotes the neighbors of the $i$-th node in the radius graph.

**Equivariant TFN Refinement Layer.** Each layer has a different set of weights for all four types of edges: ligand-to-ligand, protein-to-protein, ligand-to-protein, and protein-to-protein. The layers first update node features before updating ligand coordinates based on them. For every edge in the graph, a message is constructed based on the invariant and equivariant features of the nodes it connects. This is done in an equivariant fashion via tensor products. The tensor product is parameterized by the edge embeddings and the invariant scalar features of nodes that are connected by the edge. To obtain a new node embedding, the messages are summed:

$$\boldsymbol{h}_i^{k+1} \leftarrow \mathbf{h}_i^k + \text{BN}\left( \frac{1}{|\mathcal{N}_i|} \sum_{j \in \mathcal{N}_i} Y(\hat{r}_{ij}) \otimes_{\psi_{ij}} \mathbf{h}_j^k \right) \tag{5}$$
$$\text{with } \psi_{ij} = \Psi(e_{ij}, \mathbf{h}_i^k, \mathbf{h}_j^k)$$

Here, BN is the (equivariant) batch normalization of the e3nn library. The orders of all features are always restricted to a maximum of 1. The neural networks $\Psi$ have separate sets of weights for all 4 kinds of edges. Using these new node features and the previous layer's ligand position update $\hat{\boldsymbol{x}}^k$ (or the input positions $\hat{\boldsymbol{x}}^0 = \boldsymbol{x}_t$ for the first layer), the next ligand position update $\hat{\boldsymbol{x}}^{k+1}$ is produced via an O(3) equivariant linear layer $\Phi$ of the e3nn library:

$$\hat{\boldsymbol{x}}^{k+1} \leftarrow \hat{\boldsymbol{x}}^{k+1} + \Phi(\boldsymbol{h}^{k+1}) \tag{6}$$

**Invariant Graph Attention Layers.** These layers are based on PIFOLD and update both node and edge features. The initial features are described in the paragraphs above. We denote these as $\boldsymbol{h}_i^l$ and $\boldsymbol{e}_{ji}^l$ for the $l$-th graph attention layer to disambiguate with the features $\boldsymbol{h}_i^k$ of the equivariant refinement TFN layers. When aggregating the features for the $i$-th node, attention weights are first created and then used to weight messages from each neighboring node. With $||$ denoting concatenation and $\Omega, \Xi,$ and $\Pi$ being feed-forward neural networks, the update is defined as:

$$\begin{aligned} w_{ji} &\leftarrow \Pi(\boldsymbol{h}_j^l || \boldsymbol{e}_{ji}^l || \boldsymbol{h}_i^l) \\ a_{ji} &\leftarrow \frac{\exp w_{ji}}{\sum_{a \in \mathcal{N}_i} \exp w_{ai}} \\ \boldsymbol{v}_j &= \Xi(\boldsymbol{e}_{ji}^l || \boldsymbol{h}_j^l) \\ \boldsymbol{h}_i^{l+1} &= \sum_{j \in \mathcal{N}_i} a_{ji} \boldsymbol{v}_j. \end{aligned} \tag{7}$$

609 We drop the *global context attention* used in PIFOLD as we did not find them to be helpful for
610 sequence recovery in any of our experiments. This was with and without ligands.

611 Based on the new node features, the edge features are updated as follows:

$$e_{ji}^{l+1} = \Omega(h_j^{l+1}||e_{ji}^l||h_i^{l+1}) \tag{8}$$

## B  Discussion

613 HARMONICFLOW has the ability to produce arbitrary bond lengths and bond angles. This distin-
614 guishes it from DIFFDOCK [Corso et al., 2023], which only changes torsion angles, translation, and
615 rotation of an initial seed conformer. Thus, unlike DIFFDOCK, HARMONICFLOW would be able to
616 produce unrealistic local structures. That this is not the case, as shown in Figure 7 attests to how
617 HARMONICFLOW learns physical constraints. Still, we argue that the role of deep learning genera-
618 tive models should be to solve the hard problem of finding the correct coarse structure. If one desires
619 a conformer with low energy with respect to some energy function, this can be easily and quickly
620 obtained by relaxing with that energy function.

## C  Additional Results

### C.1  Docking without residue idenitities

Table 5: **HARMONICFLOW vs. DIFFDOCK DIFFUSION without residue idenitites.** Comparison
on PDBBind splits for docking without residue identities into *Distance-Pockets* (residues close to
ligand) and *Radius-Pockets* (residues within a radius of the pocket center). The columns "%<2"
show the fraction of predictions with an RMSD to the ground truth that is less than 2Å (higher is
better). "Med." is the median RMSD (lower is better). *These runs do not yet use self-conditioning.

| | Sequence Similarity Split | | | | Time Split | | | |
| | Distance-Pocket | | Radius-Pocket | | Distance-Pocket | | Radius-Pocket | |
| Method | %<2 | Med. | %<2 | Med. | %<2 | Med. | %<2 | Med. |
|---|---|---|---|---|---|---|---|---|
| DIFFDOCK DIFFUSION | 27.1 | 3.2 | 14.3 | 4.3 | 22.5 | 3.6 | 12.5 | 4.8 |
| HARMONICFLOW | 29.9 | 3.0 | 19.2* | 3.4* | 31.5* | 3.0* | 29.2* | 3.2* |

623 For our binding site design, it is important that the structure modeling of the ligand is accurate given
624 the evidence that having a good model of the (multi-)ligand structure is important for recovering
625 pockets and given the interlink between 3D structure and binding affinity / binding free energy. In
626 the main text Section 4.1, we investigated HARMONICFLOW's performance for docking with known
627 residue identities. However, when using HARMONICFLOW for binding site design, the residue
628 identities are not known a prior, and structure reasoning abilities in this scenario are required.

### C.2  Blind Docking

Table 6: **HARMONICFLOW vs. DIFFDOCK DIFFUSION for blind docking.** Comparison on
PDBBind splits for blind docking where the binding pocket of the protein is not known, and the
whole protein is given as input. The columns "%<2" show the fraction of predictions with an
RMSD to the ground truth that is less than 2Å (higher is better). "Med." is the median RMSD in Å
(lower is better).

| | Sequence Split | | | Time Split | | |
| Method | %<2 | %<5 | Med. | %<2 | %<5 | Med. |
|---|---|---|---|---|---|---|
| DIFFDOCK DIFFUSION | 10.7 | 40.6 | 5.9 | 12.6 | 44.1 | 5.6 |
| HARMONICFLOW | 10.1 | 41.9 | 5.8 | 20.4 | 49.4 | 5.0 |

In blind docking, the binding site/pocket of the protein is unknown, and the task is to predict the binding structure given the whole protein. While in, e.g., drug discovery efforts and in our binding site design task, the pocket is known, many important applications exist where discovering the binding site is necessary. In these experiments, the runs take longer to converge than in the pocket-level experiments. Thus, the DiffDock runs were trained for 500 epochs while the HARMONICFLOW runs were trained for 250 epochs instead of the 150 epochs in the pocket-level experiments. Table 6, shows that HarmonicFlow is also bett

### C.3 Predicted Complex Visualizations

We visualize generated structures of HARMONICFLOW in Figure 7 from the PDBBind test set under the time-based split of Stärk et al. [2022] in which there are no ligands whose SMILES string was already in the training data. The generated complexes show very chemically plausible ligand structures even though there are no local structure constraints as in DIFFDOCK and HARMONICFLOW has full flexibility in modeling bond angles and bond lengths.

In Table 5, we provide the docking results without residue identities and find that HARMOICFLOW substantially outperforms DIFFDOCK's diffusion generative process, justifying HARMONICFLOW's use in FLOWSITE for binding site design.

## D  Additional Related Work

### D.1  Flow Matching, Stochastic Interpolants, and Schrodinger Bridges

While our exposition of flow matching in the main text focused on the works of Lipman et al. [2022] and Tong et al. [2023b], the innovations in this field were made by multiple papers concurrently. Namely, Action Matching [Neklyudov et al., 2023], stochastic interpolants [Albergo & Vanden-Eijnden, 2022], and rectified flow [Liu et al., 2022] also proposed procedures for learning flows between arbitrary start and end distributions.

An improvement to learning such flows would be if their transport additionally performs the optimal transport between the two distributions with respect to some cost. With shorter paths with respect to the cost  metric, even fewer integration steps can be performed, and integration errors are smaller. Towards this, Tong et al. [2023b] and Pooladian et al. [2023] concurrently propose mini-batch OT where they train with conditional flow matching but define the conditional paths between the optimal transport solution within a minibatch. They show that in the limit of the batch size, the flow will learn the optimal coupling.

This can be extended to learning Schrodinger bridges in a simulation-free manner [Tong et al., 2023a] by learning both a flow and a score or via an iterative flow-matching and coupling definition procedure [Shi et al., 2023] akin to rectified flows. Simulation-free here means that the learned vector fields no longer need to be rolled out / simulated during training, which is memory and time-consuming and prohibits learning Schrodinger bridges for larger applications. This was required for previous procedures for learning Schrodinger bridges [Bortoli et al., 2023; Chen et al., 2022].

### D.2  Antibody Design

Another domain where joint sequence and structure design has already been heavily leveraged is antibody design [Jin et al., 2022; Verma et al., 2023; Martinkus et al., 2023]. In this task, the goal is to determine the residue types of the complementary determining regions/loops of an antibody to bind an epitope. These epitopes are proteins, and we have the opportunity to leverage evolutionary information. A modeling approach here only has to learn the interactions with the 20 possible amino acids that the epitope is built out of. Meanwhile, in our design task, where we wish to bind arbitrary small molecules, we are faced with a much wider set of possibilities for the ligand.

### D.3  Small molecule design

Another frontier where designing structure and "2D" information simultaneously has found application is in molecule generation. For instance, Vignac et al. [2023a] and Vignac et al. [2023b] show how a joint diffusion process over a small molecule's positions and its atom types can be used to

successfully generate novel realistic molecules. This task was initially tackled by EDM [Hooge-boom et al., 2022] and recently was used to benchmark diffusion models with changing numbers of dimensions [Campbell et al., 2023].

Often, it is relevant to generate molecules conditioned on context. In particular, a highly valuable application, if it works well enough, would be generating molecules conditioned on a protein pocket to bind to that pocket [Lin et al., 2022; Schneuing et al., 2023]. These applications would be most prominent in the drug discovery industry, where the first step in many drug design campaigns is often to find a molecule that binds to a particular target protein that is known to be relevant for a disease. In our work with FLOWSITE, we consider the opposite task where the small molecule is already given, and we instead want to design a pocket to bind this molecule. Here, the applications range from enzyme design (for which the first step of catalysis is binding the reactants [Nelson & Cox, 2004]) over antidote design to producing new biomedical marker proteins for use in medicinal diagnosis and biology research.

### D.4 Protein-Ligand Docking

Historically, docking was performed with search-based methods [Trott & Olson, 2010; Halgren et al., 2004; Thomsen & Christensen, 2006] that have a scoring function and a search algorithm. The search algorithm would start with an initial random conformer and explore the energy land-scape defined by the scoring function before returning the best scoring pose as the final prediction. Recently, such scoring functions have been parameterized with machine learning approaches [Mc-Nutt et al., 2021; Méndez-Lucio et al., 2021b]. In these traditional docking methods, to the best of our knowledge, only extensions of Autodock Vina [Trott & Olson, 2010] support multiligand docking. However, this still requires knowledge of the complete sidechains, which is not available in our binding site design scenario.

## E  Experimental Setup Details

In this section, we provide additional details on how our experiments were run next to the exact commands and code to reproduce the results available at `https://anonymous.4open.science/r/wolf`. In all of the paper, we only consider heavy atoms (no hydrogens).

**Training Details.** For optimization, we use the Adam optimizer [Kingma & Ba, 2014] with a learning rate of 0.001 for all experiments. The batch size for pure structure prediction experiments is 4, while that for binding site recovery experiments is 16. To choose the best model out of all training epochs, we run inference every epoch for experiments that do not involve structure modeling and every 5 epochs for the ones that do. The model that is used for the test set is the one with the best metric in terms of sequence recovery or fraction of predictions with an RMSD below 2 Å. When training for binding site recovery, we limit the number of heavy atoms in the ligand to 60. We note that for the structure prediction experiments for Binding MOAD in Table 2, the dataset construction for both methods had a mistake where ligands were selected based on their residue ID, which is incorrect because a ligand in a different chain could have the same residue ID - we will correct this in the next version of the manuscript. All models were trained on a single A100 GPU. The models that involve structure prediction were trained for 150 epochs, while those without structure modeling and pure sequence prediction converge much faster in terms of their validation metrics and are only trained for 50 epochs. The DIFFDOCK models are all trained for 500 epochs.

**Hyperparameters.** We tuned hyperparameters on small-scale experiments in the Distance-Pocket setup for HARMONICFLOW and transferred these parameters to FLOWSITE, whose additional parameters we tested separately. The tuning for both methods was light, and we mainly stuck with the initial settings that we already found to work well. By default, our conditional probability path $p_t(\boldsymbol{x}|\boldsymbol{x}_0, \boldsymbol{x}_1) = \mathcal{N}(\boldsymbol{x}|t\boldsymbol{x}_1 + (1-t)\boldsymbol{x}_0, \sigma^2)$ uses $\sigma = 0.5$ for which we also tried $0.1, 0.3, 0.5, 0.8$. The number of integration steps we use is 20 for all methods, including EIGENFOLD DIFFUSION and DIFFDOCK DIFFUSION. The number of scalar features we use is 32, and we have 8 vector features and 6 of our equivariant refinement TFN layers.

**DIFFDOCK DIFFUSION baseline.** This only uses the score model, the diffusion generative model component of DIFFDOCK [Corso et al., 2023]. We do not use the confidence model, which is a significant part of their docking pipeline. Such a discriminator could also be used on top of

HARMONICFLOW, and here, we only aim to compare the generative models. For this, we use the code at https://github.com/gcorso/DiffDock to train DIFFDOCK with our pocket definitions using the same number of scalar features and vector features using 5 of its default TFN layers followed by its pseudo torque convolution and center-convolution. We train all experiments with DIFFDOCK for 500 epochs.

**EIGENFOLD DIFFUSION baseline.** Here, we use an identical architecture as for HARMONICFLOW and only replace the flow matching training and inference with the diffusion training and inference approach of EIGENFOLD [Jing et al., 2023]. The models were trained in the same settings, and most parameters that we use in HARMONICFLOW were first optimized with EIGENFOLD DIFFUSION since we used it initially.

# F   Dataset Details

We use **PDBBind** version 2020 with 19k complexes to evaluate the structure generation capability of flow matching and the ability of FLOWSITE to design binders for a single connected ligand. We employ two dataset splits. The first is based on time, which has been heavily used in the DL community [Stärk et al., 2022; Corso et al., 2023]. The second is sequence-based with a maximum of 30% chain-wise similarity between train, validation, and test data. Buttenschoen et al. [2023] found DL docking methods to be significantly more challenged by sequence similarity splits.

For many binding pocket design tasks, it is required to bind multi-ligands. For example, when designing enzymes for multiple reactants. Such multi-ligands are present in **Binding MOAD**. We use its 41k complexes with a 30% sequence similarity split carried out as described above. We construct our *multi-ligands* as all molecules and ions that have atoms within 4Å of each other. An example of an enzyme with all substrates in the pocket as multi-ligand is in Figure 1.

## F.1   PDBBind

We use PDBBind dataset [Liu et al., 2017] with protein-ligand complexes of high binding affinity extracted and hand curated from the Protein Data Bank (PDB) [Berman et al., 2003]. For this, we use two splits.

**Splits.** Firstly, the time split proposed by Stärk et al. [2022], which now is commonly used in the machine learning literature when benchmarking docking approaches, although Buttenschoen et al. [2023] among others found many shortcomings of this split, especially for blind docking. Chiefly among them is the fact that of the 363 test complexes, only 144 are not already included in the training data if a protein is counted the same based on UniProtID. The split has 17k complexes from 2018 or earlier for training/validation, and the mentioned 363 test samples are from 2019. Additionally, there is no ligand overlap with the training complexes based on SMILES identity. The data can be downloaded from https://zenodo.org/record/6408497 as preprocessed by These files were preprocessed by Stärk et al. [2022] with Open Babel before "correcting" hydrogens and flipping histidines with by running reduce https://github.com/rlabduke/reduce. For benchmarking traditional docking software, this preprocessed data should not be employed since the hydrogen bond lengths are incorrect. For our deep learning approaches that only consider heavy atoms, this is not relevant.

Secondly, a sequence similarity, which Buttenschoen et al. [2023] found to be a more difficult split than the time split for the blind docking scenario. To create this split, we cluster each chain of every protein with 30% sequence similarity. The clusters for training, validation, and test are then chosen such that each protein's chains have at least 30% sequence similarity with any other chain in another part of the split. This way, we obtain 17741 train, 688 validation, and 469 test complexes. After filtering for complexes that have at least one contact (a protein residue with a heavy atom within 4Å), 17714 train complexes remain while no validation or test complexes are filtered out.

**Dataset Statistics.** In Figure 8, we show the number of atoms per ligand in two histograms, while Figure 9 shows the number of contacts (a protein residue with a heavy atom within 4Å) per ligand. These statistics are for the training data.

### F.2    Binding MOAD Dataset

**Split.** The sequence similarity split that we use for BindingMOAD is carried out equivalently as for PDBBind described in Section F.1. This way, we obtain 56649 of Binding MOAD's biounits for training, 1136 for validation, and 1288 as the test set. We discarded some of the biounits and only ended up with 54575 of them since 2.1k of them did not contain any other atoms besides protein atoms and waters. From these, we only use the complexes denoted as the first biounit to reduce redundancy and have only one biounit per PDB ID after which 38477 training complexes remain. We further filter out all ligands that have only one contact (a protein residue with a heavy atom within 4Å) with their protein to obtain 36203 train, 734 validation, and 756 test proteins with a unique PDB ID for each of them.

**Dataset Statistics.** Here, we provide statistics for the Binding MOAD training data. In Figure 10, we show the number of ligands per protein that is obtained under our definition of ligands and multi-ligands. Each ligand in the depicted histogram can either be a multi-ligand or a single molecule. Each multi-ligand is only counted once. In Figure 11, we show the number of atoms per ligand in two histograms, while Figure 12 shows the number of contacts per ligand.

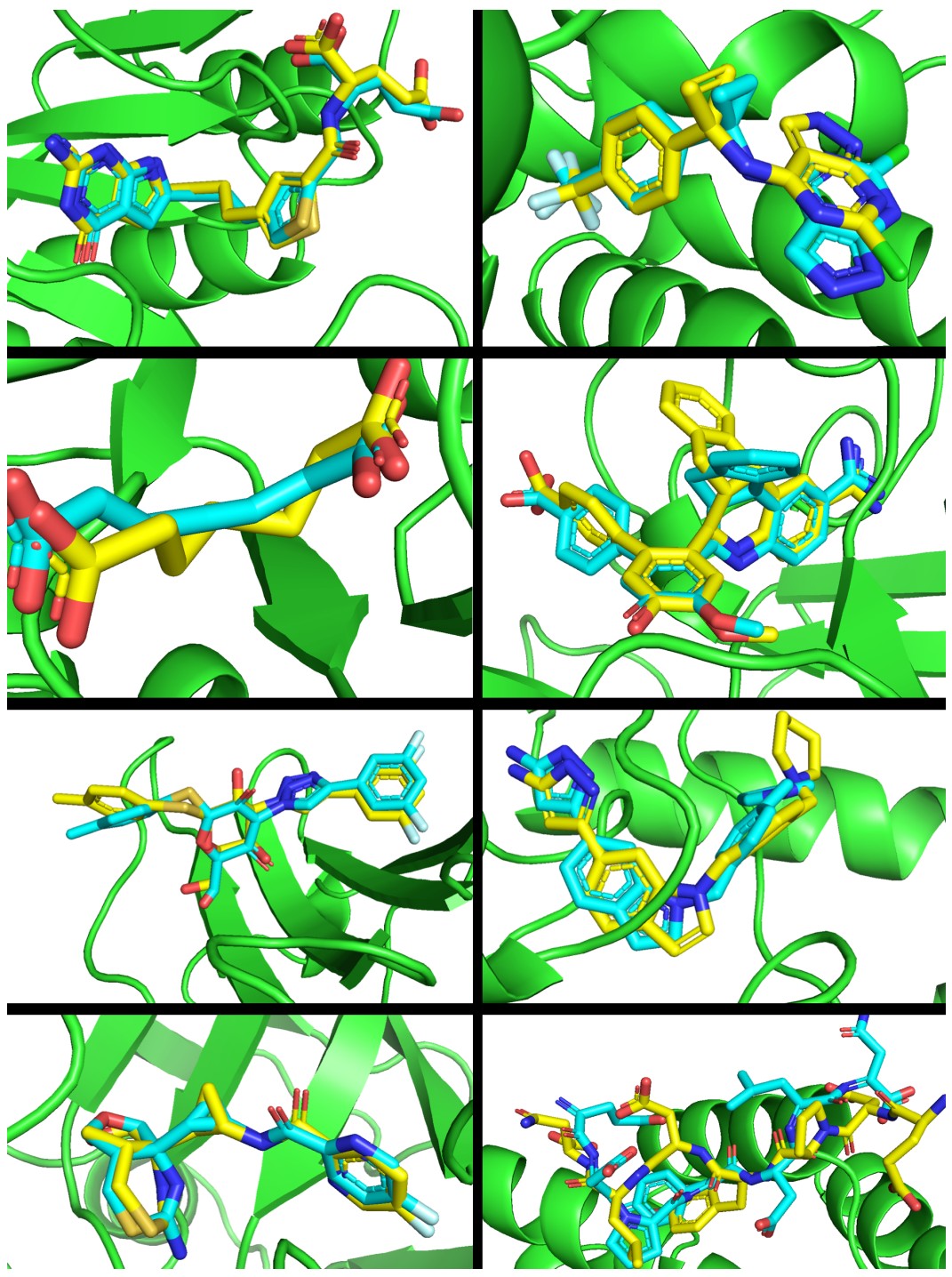

Figure 7: **HARMONICFLOW generated complexes.** Generated complexes of HARMONICFLOW for eight randomly chosen complexes in the PDBBind test set in the Distance-Pocket setup with a time-split where none of the ligands were seen during training.

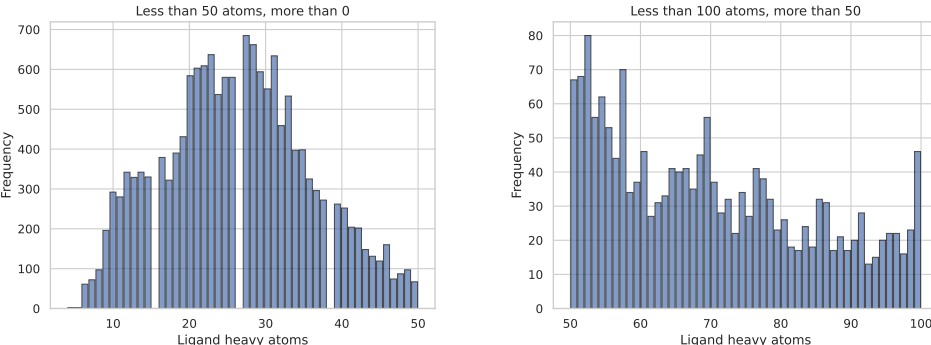

Figure 8: **Number of atoms per ligand: PDBBind.** Histograms showing the number of heavy atoms for all ligands under our ligand definition. This includes many ions, which can be important to filter out if not relevant to the desired application.

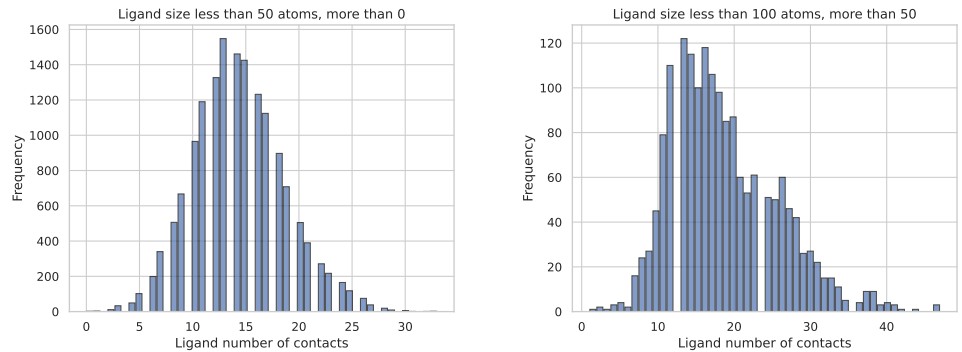

Figure 9: **Number of protein contacts per Ligand: PDBBind.** Histograms showing the number of contacts that each ligand has with its protein. A contact is defined as having a residue with a heavy atom within 4A of any ligand heavy atom.

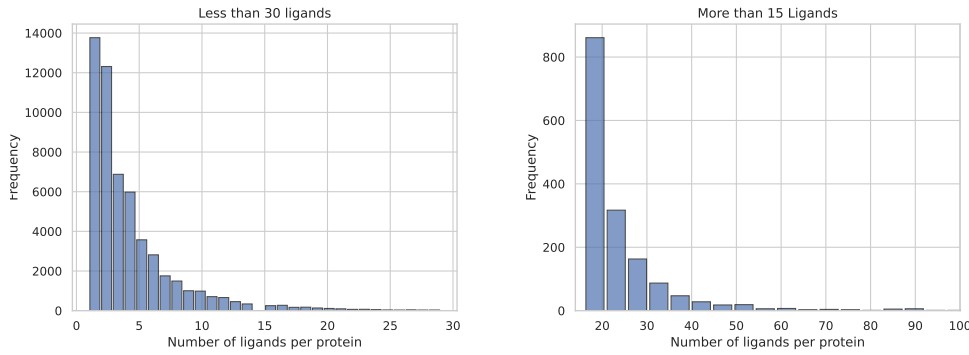

Figure 10: **Number of Ligands per Protein: Binding MOAD.** Histograms showing the number of (multi-)ligands per protein in the Binding MOAD dataset under our ligand definition. Each ligand here can be a multi-ligand. In that case, it is only counted once.

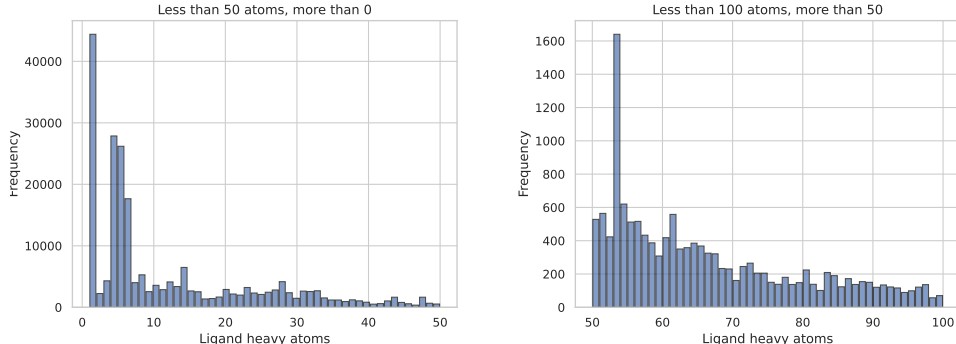

Figure 11: **Number of atoms per ligand: Binding MOAD.** Histograms showing the number of heavy atoms for all ligands under our ligand definition. This includes many ions, which can be important to filter out if not relevant to the desired application.

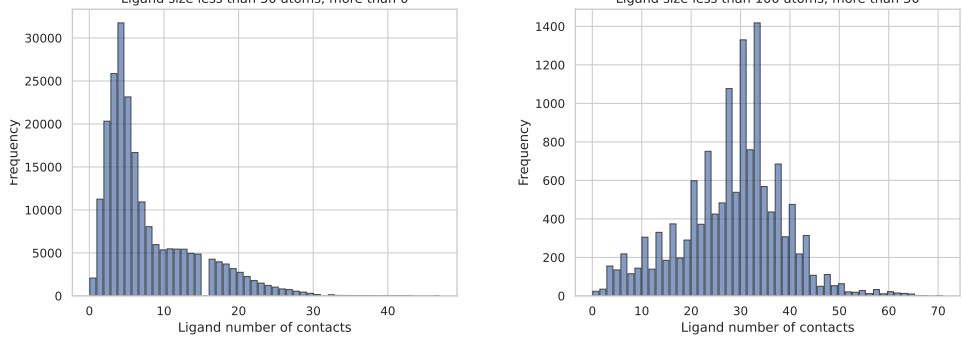

Figure 12: **Number of protein contacts per Ligand: Binding MOAD.** Histograms showing the number of contacts that each ligand has with its protein. A contact is defined as having a residue with a heavy atom within 4A of any ligand-heavy atom.

