# OpenReview forum: "Harmonic Prior Self-conditioned Flow Matching for Multi-Ligand Docking and Binding Site Design"
_NeurIPS.cc/2023/Workshop/AI4Science — NeurIPS2023-AI4Science Oral_

### Official Review · Reviewer_pWwh · 2023-10-21
**Interesting Technical Innovations, weaker evaluation**

**Rating:** 7
**Confidence:** 4

**Review:**

1. Overall evaluation
Opinion: The authors present a novel method based on flow matching to perform protein-ligand docking and binding site design. Their method combines several interesting components like TFN layers used in a novel way, harmonic priors, and self-conditioning. While the authors claim SOTA on generative processes for docking, they do not compare to strong physics-based baselines, limiting the scope of their claims. Overall, I think the paper is an interesting new take at both docking and binding site generation but should work on the evaluation and relation to previous work. I vote accept.

Summary: The authors propose a model called HarmonicFlow which is based on the flow matching objective. They compare its performance to DiffDock and show that it performs better. They then extend it to also designing the protein pocket via discrete residue type generation and evaluate its performance on this task via sequence recovery and BLOSUM score.

Contributions:
[C1] they show that their model can be used for multi-ligand docking and outperforms previous deep learning methods.
[C2] they introduce some technical innovations such as a new way to use TFN layers.

2. Strengths & Weaknesses:
[S1] Interesting results on sequence similarity split: compared to previous work that just performed time-based splits, this work looks at the biologically rigorous sequence similarity split. Interestingly, while the authors call this split harder, they seem to be getting better performance on it than on the time-based split.
[S2] Good use case for flow matching: The advantage of flow matching methods over vanilla DDPMs of choosing your prior is nicely leveraged here by defining the prior as the ligands and target protein are separated.

[W1] Selection of baselines: DiffDock has previously been shown to be outperformed by simple physics-based docking programs if these are used correctly. Therefore, a comparison to these methods like GLIDE would have been illuminating.
[W2] enzyme design claims: the authors claim that this work opens up enzyme design possibilities. However, already in the 80s/90s people tried to design enzymes based on substrate binding and found that this stabilisation of the ground state actually diminishes function, so this claim should be a bit tuned down.

---

### Official Review · Reviewer_ePhU · 2023-10-24
**New SOTA for ligand docking and binding site design**

**Rating:** 8
**Confidence:** 4

**Review:**

The paper presents flow matching algorithm with harmonic prior for ligand docking and binding site design. The method surpasses previous SOTA method DiffDock in terms of ligand RMSD between ground truth and generation. Overall this a very interesting and well-written work and worth attention from a broad AI4SCI community.

The paper use EigenFold Diffusion as a baseline to compare with, but EigenFold was designed to generate conformation ensembles. It would be better if the authors can articulate more on the implementation of EigenFold Diffusion.

---

### Meta-Review · Area_Chair_Dqsz · 2023-10-26

**Recommendation:** Accept (Oral)
**Confidence:** 3

**Metareview:**

This work introduces an innovative flow-based generative model that constructs binding pockets for ligands from their 2D chemical graph and 3D coordinates representing the protein pocket's backbone atomic structure. The ligand coordinates are initialized using a harmonic prior. The generative process uses an ODE, defined by a flow model for iterative updates on residue types and 3D ligand coordinates. The effectiveness and performance are well validated by extensive experiments on the PDBBind and Binding MOAD datasets. However, some equations are inherently complex and it is often hard to follow this paper. The authors assume that readers have a deep understanding of some specific mathematical concepts. The absence of a background on these concepts makes comprehension harder.